# Long-read sequencing transcriptome quantification with lr-kallisto

Rebekah K. Loving[1], Delaney K. Sullivan[1,2], Fairlie Reese[3,4], Elisabeth Rebboah[3,4], Jasmine Sakr[3,4], Narges Rezaie[3,4], Heidi Y. Liang[3,4], Ghassan Filimban[3,4], Shimako Kawauchi[3], A. Sina Booeshaghi[5], Páll Melsted[6,7], Conrad Oakes[1], Diane Trout[1], Brian A. Williams[1], Grant R. MacGregor[3], Barbara J. Wold[1]*, Ali Mortazavi[3,4]*, Lior Pachter[1,8]*

1 Division of Biology and Biological Engineering, California Institute of Technology, Pasadena, California, United States of America, 2 UCLA-Caltech Medical Scientist Training Program, David Geffen School of Medicine, University of California, Los Angeles, Los Angeles, California, United States of America, 3 Developmental and Cell Biology, University of California Irvine, Irvine, California, United States of America, 4 Center for Complex Biological Systems, University of California Irvine, Irvine, California, United States of America, 5 Department of Bioengineering, University of California, Berkeley, Berkeley, California, United States of America, 6 deCODE Genetics/Amgen Inc., Sturlugata Reykjavík, Iceland, 7 Faculty of Industrial Engineering, Mechanical Engineering and Computer Science, School of Engineering and Natural Sciences, University of Iceland, Sæmundargata Reykjavík, Iceland, 8 Department of Computing and Mathematical Sciences, California Institute of Technology, Pasadena, California, United States of America

* woldb@caltech.edu (BJW); ali.mortazavi@uci.edu (AM); lpachter@caltech.edu (LP)

## Abstract

RNA abundance quantification has become routine and affordable thanks to high-throughput "short-read" technologies that provide accurate molecule counts at the gene level. Similarly accurate and affordable quantification of definitive full-length, transcript isoforms has remained a stubborn challenge, despite its obvious biological significance across a wide range of problems. "Long-read" sequencing platforms now produce data-types that can, in principle, drive routine definitive isoform quantification. However some particulars of contemporary long-read datatypes, together with isoform complexity and genetic variation, present bioinformatic challenges. We show here, using ONT data, that fast and accurate quantification of long-read data is possible and that it is improved by exome capture. To perform quantifications we developed lr-kallisto, which adapts the kallisto bulk and single-cell RNA-seq quantification methods for long-read technologies.

## Author summary

Long-read sequencing technologies have become essential for genomics, specifically for genome assembly applications, where they help to resolve complex repetitive regions. However, despite their widespread use for assembly, long-read sequencing technologies have not been used much for quantification of

**Data availability statement:** The LRGASP data can be accessed from the accessions and ftp links listed in the data folder of https://github.com/pachterlab/LSRRSRLFKOTWMWMP_2024. IGVF Bridge exome capture and non-exome capture can be accessed from the IGVF portal with the accession IDs in the provided table. The HCT116 cell line SG-NEx data was accessed on March 13, 2024 at https://registry.opendata.aws/sg-nex-data. The lr-kallisto method is available via release 0.51 of kallisto at https://github.com/pachterlab/kallisto. IGVF Bridge exome capture and nonexome capture data provided in format (Accession IDs, Subpool Name, Read Type): (IGVFDS4803WKTQ, B01_13G, Nanopore); (IGVFDS9445YYVB, B01_13H, Nanopore); (IGVFDS9522BMQK, B01_13G, Illumina); (IGVFDS0356VCIO, B01_13H, Illumina). We used bambu v3.4.1, IsoQuant v3.3.0, and oarfish v0.5.1 (with the exception of analysis of HCT116 data). In the initial version of the preprint, oarfish (v0.3.1 and v0.4.0) were used and the simulation data was run with samtools sort (genome coordinate sorting), causing overcounting in oarfish's performance due to oarfish's use of consecutive alignments of the same read filtering; this has been updated in this version of the manuscript. Simulation data is available at https://zenodo.org/records/11201284. Processed abundance matrices for Figs 1–3 are available at https://zenodo.org/records/13755772. Code for reproducing the results and figures in the manuscript is available at https://github.com/pachterlab/LSRRSRLFKOTWMWMP_2024.

**Funding:** R.K.L. was funded by United States Department of Energy, Office of Science, Office of Advanced Scientific Computing Research, Department of Energy Computational Science Graduate Fellowship under Award Number DE-SC0020347. F.R., E.R., J.S., N.R., H.Y.L., G.F., S.K., A.S.B., D.T., B.A.W., G.M., B.J.W., A.M., and L.P. were partially supported by NIH UM1 HG012077. D.K.S. was supported by the UCLA-Caltech Medical Scientist Training Program (NIH NIGMS training grant T32 GM008042).

genes and transcripts in RNA-seq experiments. Initially, this was primarily due to cost: the extent of sequencing needed for accurate quantification was prohibitive. While costs have decreased in recent years, another obstacle remained, namely the development of efficient computational methods that could accurately associate long-reads with their transcripts of origin. In this work we describe a novel algorithm for long-read quantification which can be used with any long-read technology. We show that an implementation of our method, which we term lr-kallisto, is accurate and efficient, making possible quantification of long-read sequencing single-cell or bulk RNA-seq libraries. This will enable more complete, accurate, and scalable assessment of RNA content in cells than has been possible with short-read sequencing technologies.

## Introduction

Advances in long-read RNA sequencing are facilitating transcript discovery, annotation improvements, and detection of isoform switching, thanks to reductions in cost and decreasing error rates as the technologies mature [1–3]. Specifically, long-read RNA-seq can readily detect gene fusion transcripts and other expressed rearrangements in cancer [4], and isoform switching of biological consequence across development [5,6]. In translational genomics, precision medicine workflows are increasingly including gene and transcript ontology. These capabilities depend, in part, on accurate annotation of the genomes and transcriptomes of human and model organisms, though they remain incomplete [7,8]. Improvements in long-read sequencing now allow for much needed refinement of annotations for human and model organisms, coupled with rapid generation of genomes and annotations for non-model organisms [9]. Importantly, while annotation is mainly facilitated by transcript discovery, quantification of isoforms is critical for filtering and thresholding steps that are prerequisites for resolving locus structure and quantifying their expression products [10].

While recent increases in affordability and sequence quality are bringing full-isoform quantification within reach, the long-read platforms are still rapidly changing and less mature than short-read technologies [2]. For example, Oxford Nanopore Technology (ONT) sequencing has evolved over many versions of chemistry in the library preparation kits, pores, and signal processing algorithms. This has resulted in a range of ONT data with various error profiles and error distributions within the sequences. Of the quantification tools that have been developed so far [11–19], many are optimized for performance with a given generation of long-read data and are now antiquated, in both accuracy and efficiency, for processing the low error rate ONT data currently being produced. Moreover, many methods are based on the assumption of near uniform distribution of sequencing error along reads; we found, as have others [20], that this does not hold in practice. Furthermore, some ONT sequencing biases have now been described, including non-uniformly distributed sequencing error and sequence influenced error, such as higher GC content and repeat regions increasing sequencing/base calling error [21].

The funders had no role in study design, data collection and analysis, decision to publish, or preparation of the manuscript.

**Competing interests:** The authors have declared that no competing interests exist.

By contrast, several accurate and efficient tools have been developed for short read RNA-seq preprocessing [22–27]. However, even with the recent significant reduction in the long-read RNA-seq error rates to ~0.5%, sequencing errors remain informatically problematic and are comparatively much higher than the ~0.01% in short-read RNA-seq. This makes the application of the fastest pseudoalignment methods [25,27] to long-reads nontrivial (Fig A in S1 Text). Our approach, which builds on kallisto [23–25,28] and which we term lr-kallisto, demonstrates the feasibility of pseudoalignment for long-reads; we show via a series of results on both biological and simulated data that lr-kallisto retains the efficiency of kallisto thanks to pseudoalignment, and is accurate on long-read data. Furthermore, we show that lr-kallisto is robust to error rates, making it suitable also for the analysis of previously published older long-read sequencing data.

## Results

To assess the accuracy of lr-kallisto with respect to data from the current Oxford Nanopore Technologies platform (see Materials and Methods) we generated a deep coverage, high fidelity dataset using long-read sequence and an Illumina short-read sequence SPLiT-Seq nuclei of the left cortices of two mouse strains as part of the IGVF consortium [29]. Specifically, 4 biological replicates (2 males and 2 females) were assayed from both C57BL6/J and CAST/EiJ mice, all at 10 weeks of age, with libraries generated with and without targeted exome capture of all mouse protein coding exons using the Twist Biosciences Mouse exome panel of 215,000 probes (Fig 1A; see Methods). Thus our exome capture transcriptome will be enriched for reads overlapping one or more coding exons in the same cell. This platform and experimental design was chosen to produce starting data with a highly relevant sequencing error profile for two very well characterized genomes whose natural genetic variation between strains is similar to that found within individual human genomes. This also sets the stage for using lr-kallisto to study natural genetic variation.

We found no effective difference in read lengths with reads generated from exome capture as opposed to non-exome capture libraries (Fig 1B), though the exome capture library showed a smaller fraction of mono-exonic reads (Fig 1C). This indicated that exome capture is an effective approach to increasing the transcriptome complexity of libraries. The Illumina and ONT sequenced libraries displayed high transcript abundance concordance after quantification with lr-kallisto (Fig 1D–1G), showing that lr-kallisto accurately quantifies transcripts from long-reads, as well as demonstrating that deeply sequenced ONT libraries are suitable for high accuracy quantification. The concordance correlation coefficients (CCC), which measure how close the ONT and Illumina quantifications are to being identical, were high for both the exome capture and non-exome capture libraries (0.95 and 0.96, respectively). Importantly, when comparing all long-reads that were subject to exome capture versus those that were not, we observed a 3-fold increase in the percentage of spliced reads aligning (Fig Bi in S1 Text). Thus, we find that

exome capture reads help overcome the limitations of RNA sampling in the nucleus, including expected reads from unspliced precursor transcripts. The effect, as others have noted [30] is to provide more informative reads to study full-length, spliced transcript isoform usage at lower cost. Furthermore, lr-kallisto outperforms Bambu [15], IsoQuant [17], and Oarfish [16] with respect to CCC, Pearson correlation, and Spearman correlation (Fig Bii in S1 Text). In particular, the lr-kallisto CCC is 0.95 versus 0.82 for the recently published Oarfish long-read quantification tool. We found that lr-kallisto also outperforms Bambu (CCC = 0.86) and IsoQuant (CCC = 0.78) (Fig Bii in S1 Text), which have previously been shown to outperform other long-read quantification methods [2,31]. In addition to being more accurate than other methods, lr-kallisto is also more computationally efficient (Fig Biii in S1 Text). Note that the dramatic difference in time scales between PacBio and ONT is due to the number of reads in the ONT datasets being much higher, in general. We further bench-marked walltime and maximum resident set size of bambu, IsoQuant, lr-kallisto, and oarfish on the exome capture data, showing that lr-kallisto is 3-10x more memory efficient than oarfish, 18-50x more memory efficent than bambu, and 15-46x more memory efficient than IsoQuant (Table A in S1 Text).

Importantly, lr-kallisto can be used for both high-throughput bulk RNA-seq as well as single-cell or single-nucleus RNA-seq datasets (S1iv-vi), and is not only faster than other tools, but also benefits from the low-memory requirements of kallisto [23,28] (Table A in S1 Text). For single-nucleus RNA-seq processing, we used splitcode [32] to extract nuclei barcodes, UMIs, and reads from the raw ONT reads and then pseudoaligned and quantified the reads with lr-kallisto (see Methods). All of the barcodes from the ONT reads that passed filtering were also found in Illumina sequenced reads (Fig Biv in S1 Text). Increased UMI depth per nucleus yields higher Spearman correlations, and given that the libraries were not fully saturated by sequencing (when downsampling the ONT reads by half, we found only 803721 out of 946087 bar-codes, a reduction of 15%), it seems likely that with increased sequencing depth short and long read correlations may improve (Fig BivI in S1 Text). To assess the observed correlations between Nanopore and Illumina, we evaluated random oligo vs 3' priming in Illumina sequenced reads and ONT sequenced reads separately, in the same fashion, finding lower correlations (majority of nuclei having a Spearman $\rho$ between 0.10 and 0.30) than between ONT sequenced reads and Illumina sequenced reads (Fig BivII-III in S1 Text).

We also examined the concordance between the exome capture and non-exome capture in both long and short reads, and found it to be only CCC = 0.88, highlighting the distortion resulting from the coupling of exome capture with a mix of 3'-end and randomly primed read sequencing that is characteristic of Parse (Fig Bv in S1 Text). We then explored the difference in 3'-end and randomly primed read sequencing by looking at the mean transcript-level expression of 3'-end (polyT) reads per cell in ONT vs Illumina, in randomly primed (randO) reads per cell in ONT vs Illumina, and when reads from the two priming methods are merged per cell. We found that randomly primed reads in a cell exhibit greater cor-relation to the corresponding randomly primed reads in Illumina in the highest mean expressed 100 transcripts in ONT than with 3'-end reads in a cell exhibit (Fig Bvi in S1 Text). We found that the expression of long non-coding RNAs (lnc-RNAs) is significantly different between the exome capture and non-exome capture with 8392 lnc-RNAs with non-exome capture expression greater than 150% of exome capture expression, which combined with the difference in 3'-end and randomly primed reads highlights the bias of 3'-end reads, neglecting the non-trivial existence of functional internally priming transcripts.

The lr-kallisto quantification results are corroborated when comparing its performance to other tools on previously pub-lished data that is less deeply sequenced. In a comparison of Illumina and ONT on the HCT116 cancer cell line dataset generated by SG-NEx [33], we found that lr-kallisto could accurately quantify isoform level expression, in performance comparisons constituting two replicates of direct cDNA and direct RNA (Fig Ci in S1 Text). The CCC performance of lr-kallisto exceeded that of Oarfish, evaluated on this data in [16]. Spearman correlations were lower overall in this dataset, indicating poor data quality, perhaps due to the lower coverage and higher sequencing error rate. We also com-pared lr-kallisto's performance on direct RNA to direct cDNA (Fig Cii in S1 Text). We also found better performance with direct cDNA versus direct dRNA in replicate4, and hypothesize that this is likely due to ~4 times the depth of coverage for replicate 4 in direct cDNA (7,656,893 reads) vs th e direct RNA replicate 4 (1,896,643 reads), whereas replicate 3 direct

 

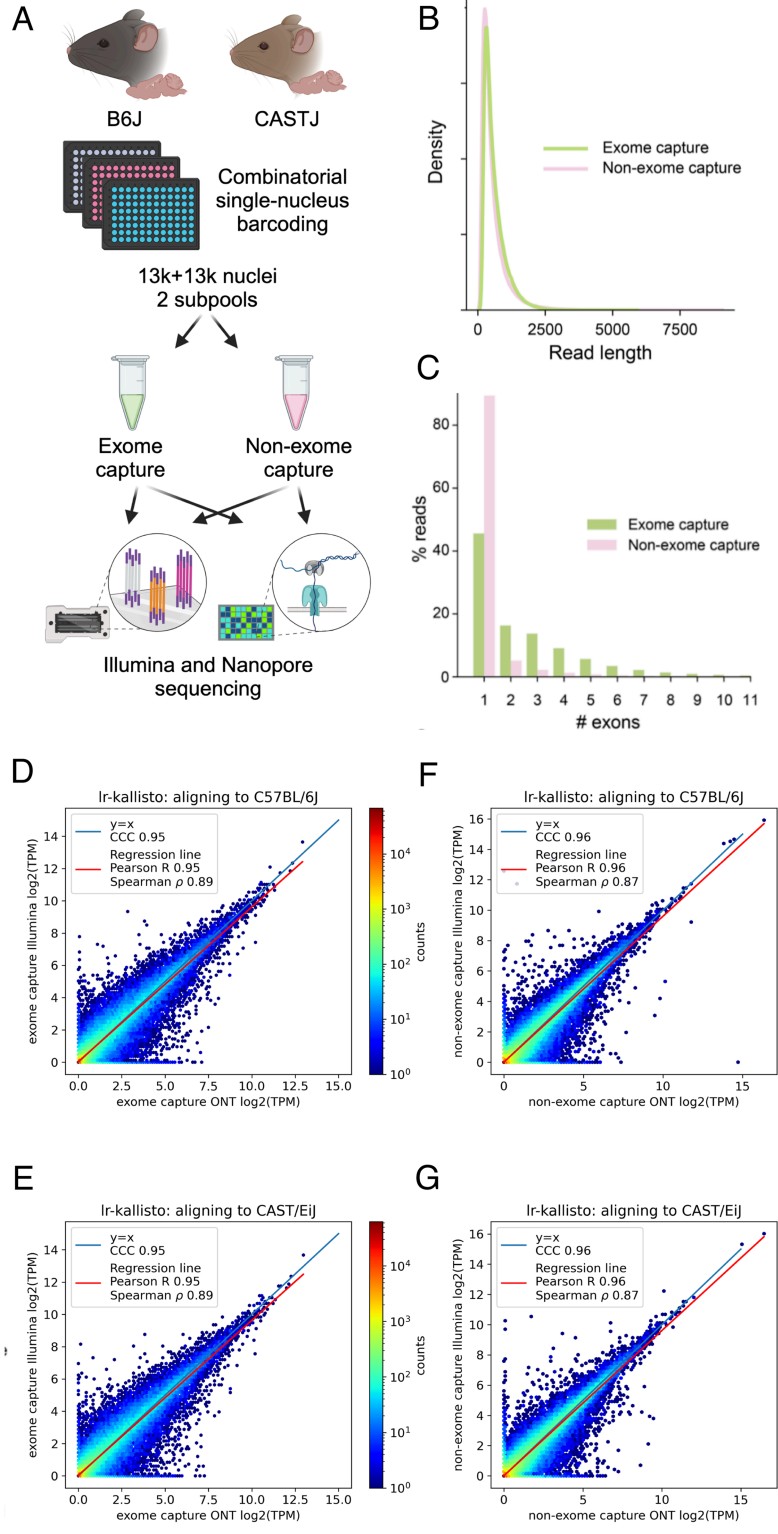

**Fig 1**. **lr-kallisto demonstrates high concordance between Illumina and ONT.** (a) Experimental overview for comparison of exome capture vs. non-exome capture LR-Split-seq libraries. (b) Kernel density estimations for read length distributions by capture strategy. (c) Percentage of demultiplexed reads by number of exons in each read between exome and non-exome capture. (d-g) Each point is a hexbin representing the number of transcript in the bin with expression in log2(TPM) with x-coordinate quantified from long reads and y-coordinate quantified from short reads. Total number of

points is the total number of annotated transcripts in the reference transcriptome. CCC is a measure of how close the data is to $x = y$, while Pearson R and Spearman $\rho$ are measures of correlation between x and y. (d) lr-kallisto pseudobulk quantifications of exome capture for the C57BL/6J sample. (e) lr-kallisto pseudobulk quantifications of exome capture for the CAST/Eij sample. (f) lr-kallisto pseudobulk quantifications of non-exome capture for the C57BL/6J sample. (g) lr-kallisto pseudobulk quantifications of non-exome capture for the CAST/Eij sample. Concordance Correlation Coefficient (CCC), Pearson, and Spearman correlations are shown for each comparison. Created with https://BioRender.com

cDNA (873,077 reads) vs direct RNA (1,185,183 reads) does not have the increased depth of coverage. We also compared lr-kallisto to Bambu, IsoQuant, and Oarfish on a previously sequenced mouse cortex PacBio dataset (Fig Ciii in S1 Text). On this dataset [34,35], which has an error rate of 12.4% (see Methods) and a different error profile with errors more uniformly distributed along transcripts, we found similar performance between programs with lr-kallisto slightly outperforming other tools in CCC. Further, since we use kallisto for the quantification of the short read Illumina data across these benchmarks, we highlight that kallisto, salmon, and other short read quantification tools are highly concordant. We demonstrate this with this dataset [34,35] as we are able to perform analogous mapping and quantification with kallisto and salmon on the paired-end Illumina data. Here we observe a slightly higher CCC (.822 vs .81, .797 vs .78, .801 vs .80) in lr-kallisto, IsoQuant, and bambu, respectively and a slightly lower CCC (.806 vs .81) in Oarfish when comparing PacBio transcript expressions to Illumina transcript expressions with kallisto vs with salmon (Fig Ciii in S1 Text). However, with Parse single-nucleus short read data, there is not a truly analogous processing in salmon for the pseudoalignment and quantification. Standard single-end bulk data significantly differs from Parse single-nucleus short read data, as it uses a combination of polyA priming and random hexamer priming. Therefore, with kallisto, we pseudobulked the counts without length normalization, as length normalization is not appropriate for this priming chemistry, yielding comparable quantifications to the single-nucleus quantifications as this strategy avoids adding methodological bias in alignment, counting, and disambiguating multi-mapping reads.

We benchmarked lr-kallisto's stability and robustness compared to other long-read quantification tools across species, platforms, and protocols, by evaluating lr-kallisto's performance, along with Bambu, IsoQuant, and Oarfish using the LRGASP's challenge 2 benchmark [2] of long-read quantification tools (Fig 2A–2b). For our benchmarking, we chose to focus on the Mouse ES data, as it had lower sequencing error rates across 3 out of the 4 protocol/platform combinations, thereby serving as the closer proxy for current long-read data. Further, we also used LRGASP Human WTC11 to enable an analysis of lr-kallisto's relative performance on SIRV Set 4, which provides a benchmarking of the ability to quantify complex and difficult transcript sets. The SIRV-Set 4 synthetic transcripts provide a useful control. This set includes the SIRV isoforms, the long SIRV isoforms, and the External RNA Controls Consortium (ERCC) transcripts. The SIRV-Set 4 isoforms include seven genes (SIRV1 through SIRV7) comprised of 69 fabricated human isoforms. SIRV1 through SIRV7 transcript isoforms were constructed to include mono-exon transcripts, single- and multi-exon skipping events, alternative start/stop sites, as well as antisense transcripts. ERCC transcripts range from 250 to 2000 bp in length, mimicking natural eukaryotic mRNAs. The long SIRVs (SIRV10 through SIRV12) includes 15 RNA transcripts of 4000 to 12,000 bp. All together this synthetic control provides a helpful benchmark across tools accuracy and sensitivity across transcript complexity within PacBio and ONT sequencing protocols and platforms. We found that Bambu, IsoQuant, lr-kallisto, and Oarfish all achieved reasonably high CCCs between replicates, both with respect to abundance estimates (Fig 2A), and variability between isoforms (Fig 2B). With the SIRV Set 4 analysis, we show both lr-kallisto and Oarfish lag in median relative difference (MRD) and outstrip across all platforms and protocols in Spearman Correlation Coefficient (SCC) compared to IsoQuant and Bambu, while for Normalized Root Mean Squared Error (NRMSE) lr-kallisto and Oarfish outperform IsoQuant and Bambu in PacBio, while underperforming IsoQuant in ONT (Fig 2C). Moreover, lr-kallisto and Oarfish are both high performing in the metrics of percent expressed transcripts (PET) across all three categories of transcript sets (SIRV, SIRV long transcripts, and ERCC), indicating higher detection accuracy even at low expression rates

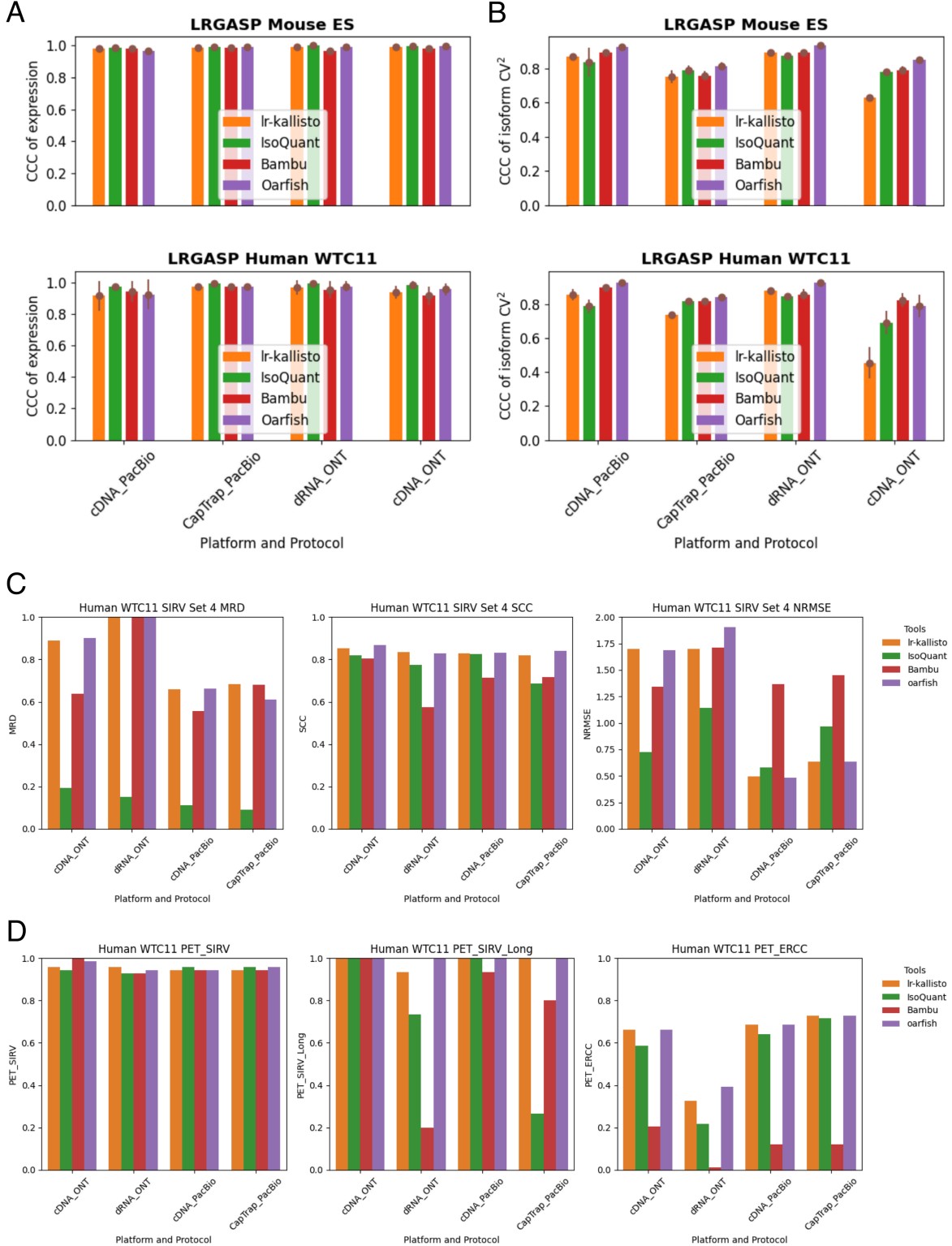

**Fig 2**. Comparison of Bambu, IsoQuant, lr-kallisto, and Oarfish in (a) abundance estimates as measured by CCC of expression and (b) variability between isoforms as measured by CCC of isoform $CV^2$, with 90% CI to measure consistency and reproducibility among replicates between the tools.

and for complex and long transcripts (Fig 2D). Furthermore, in a sequencing error free simulation with uniform expression of SIRV Set 4, we found that lr-kallisto detected all isoforms (with the exception of SIRV311, one of the mono-exonic isoforms, while maintaining detection of the other three mono-exonic isoforms: SIRV206, SIRV512, SIRV618) and was perfectly accurate in quantifying 88% of SIRV Set 4 isoforms. For completeness, we also compared the performance of lr-kallisto to Bambu, IsoQuant, Oarfish using the metrics of the LRGASP paper (Fig D in S1 Text). Resolution Entropy (RE) is a measure of how well a tool uniformly quantifies at different expression levels. Irreproducibility Measure (IM) is a measure of how reproducibly the tool quantifies expression across replicates, i.e., whether the coefficient of variation between replicates is low. Consistency Measure (CM) is a measure of how consistent the tool is at detecting expressed transcripts, assuming that transcripts should be expressed simultaneously across replicates, and ACVC is the Area under the Coefficient of Variation Curve, which again assumes that for a given mean expression level across replicates the coefficient of variation should be low. We found that lr-kallisto performs as well as other programs on these stability and robustness measures. The variability that we found in quantifications of replicates can be explained by variable depth of sequencing between the replicates and between the protocols and platforms [2]. The notable difference in ONT cDNA is due in part to a sequencing error rate of ∼12%, which is characteristic of data obtained on earlier ONT platform versions, as these were [36].

We also assessed the performance of lr-kallisto using simulated data across a range of sequencing error profiles, and compared with results on the same simulated data for five other widely used or recently published programs. We used simulations generated by [17] who used the IsoSeqSim simulator (see Data and Code Availability) to generate PacBio reads (6 million *Mus musculus* reads with ∼1.6% sequencing error rate), and NanoSim [18] to generate ONT.R10.4 reads (30 million *Mus musculus* reads with ∼2.8% sequencing error rate). The PacBio IsoSeqSim Simulation (Fig 3A) demonstrates lr-kallisto's high accuracy compared to the currently leading benchmarked long-read quantification tools Bambu, IsoQuant, and Oarfish, with lr-kallisto achieving a CCC of 0.98, vs 0.90, 0.91, and 0.99, respectively [2,31]. Furthermore, in the ONT NanoSim R10.4 Simulation (Fig 3B), lr-kallisto ties for the highest CCC of 0.97, vs 0.88 and 0.91, respectively.

We performed additional comparative evaluations of Bambu, IsoQuant, lr-kallisto, and Oarfish on a more extensive set of simulations to understand the strengths and weaknesses these tools when confronted with different sequencing error challenges (Fig E in S1 Text). We found that lr-kallisto and IsoQuant were both robust to indel and substitution profiles simulated to match PacBio sequencing data and uniformly distributed. IsoQuant was also robust to uniformly distributed sequencing errors with indel and substitution profiles matched to ONT, whereas lr-kallisto performance degraded at higher ONT error rates in this simulation (Fig Ea in S1 Text). In particular, this highlights lr-kallisto's sensitivity to the unrealistic combination of uniform sequencing error distribution and higher rate of insertion errors in ONT versus PacBio.

In another ONT simulation generated with NanoSim to produce reads with an 11.2% error rate (see Data and Code Availability), lr-kallisto achieved a CCC of 0.31 on all transcripts, outperforming IsoQuant (CCC = 0.28), and underperforming Bambu (CCC = 0.51), and Oarfish (CCC = 0.55) (Fig Eb in S1 Text). This was also the case at a higher error rate (15.2%), with lr-kallisto continuing to outperform IsoQuant and underperform Bambu and Oarfish (Bambu CCC = 0.53, IsoQuant CCC = 0.32, lr-kallisto CCC = 0.34, Oarfish CCC = 0.58) (Fig Ec in S1 Text).

The performance of lr-kallisto benefits from quantification with respect to a de Bruijn graph [28]. We tested whether and to what extent changing the *k*-mer length default in lr-kallisto to 63 bp long vs 31 bp long in the reference transcriptome de Bruijn graph creates a less connected and less complex structure (Fig F in S1 Text). In this example, of the Pax2 gene, we find that a change of *k*-mer length simplifies the T-DBG with the reduction of a single node and 2 edges. However, when we scale this out to just the first 1000 transcripts listed in the LRGASP basic gencode human annotation, we found a reduction from 3,698 nodes using the 31 *k*-mer T-DBG to 2,708 nodes using the 63 *k*-mer T-DBG and 4,687 edges to 3,238 edges, respectively. Furthermore, the largest connected T-DBG graph component in the 63 *k*-mer T-DBG is composed of 12.59% of the bp vs 65.90% in the 31 *k*-mer T-DBG. We believe that the selection of higher quality, low

## A  PacBio IsoSeqSim Simulation

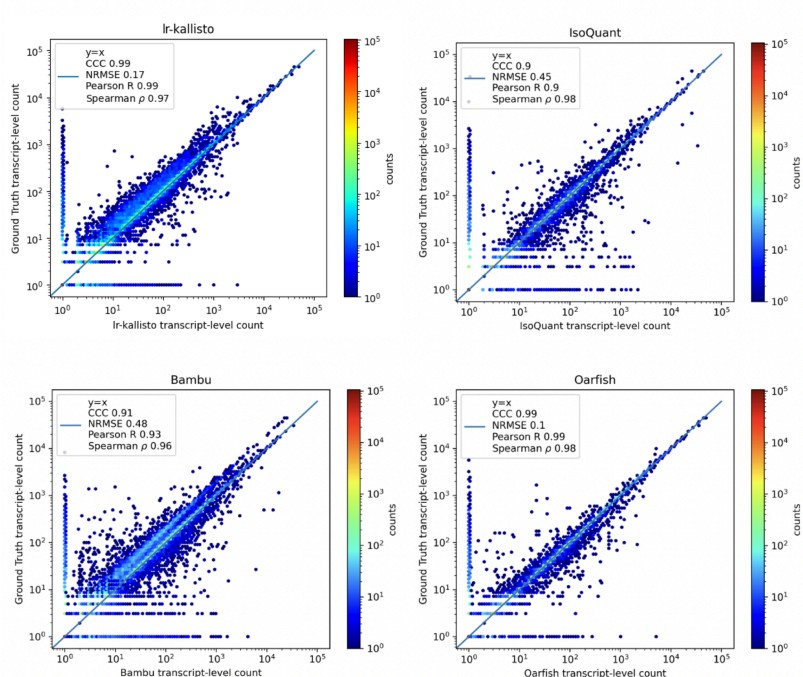

## B  ONT NanoSim R10.4 Simulation

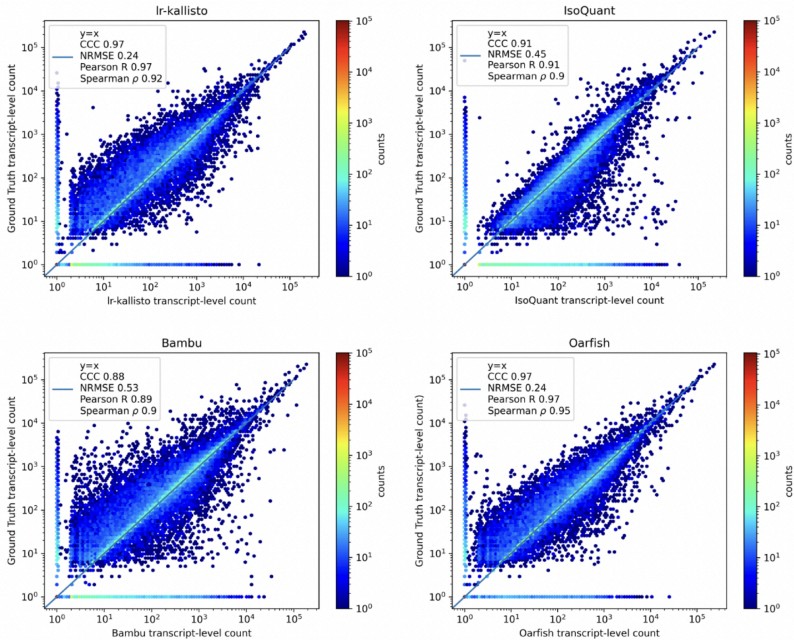

**Fig 3. lr-kallisto is highly accurate in simulations with error up to ~3%.** A comparison of performance of Bambu, IsoQuant, lr-kallisto, and Oarfish on PacBio (top) and ONT (bottom) simulations with Concordance Correlation Coefficient (CCC), Normalized Root Mean Squared Error, and Pearson's and Spearman's correlation coefficients reported.

sequencing error regions from the reads by the 63 $k$-mer T-DBG, combined increasing the probability of uniquely mapping, or at the very least mapping to a transcript compatibility class with less transcripts, is producing more accurate and more efficient pseudoalignment.

## Discussion

With Oxford Nanopore sequencing becoming more accessible due to low entry costs and reduced sequencing error rate [37], long-read sequencing is advancing the ability to routinely decipher the isoform complexity of transcriptomes. Increasing throughput now makes it possible to not only perform discovery with long-read sequencing, but also to accurately quantify transcript abundances, and we have shown that results comparable to short-read sequencing can be achieved at reasonable cost with exome capture, and with high accuracy quantification using lr-kallisto. Exome capture will be especially helpful for filtering out intronic reads that would be otherwise sequenced in (single-)nucleus data, as nuclei are replete with intron lariats and partially processed transcripts. lr-kallisto is highly accurate in producing quantification results on data with less than 10% sequencing error rate comparable to those with short-read sequencing. This makes lr-kallisto immediately useful for current long-read sequencing transcriptome projects, although performance will not be as good on legacy higher error long-read sequencing datasets. While standard kallisto is now useful and competitive with current long read tools for quantification on the high accuracy (>99.5%) long reads, demonstrating the suitability of pseudoalignment to long reads, lr-kallisto eclipses kallisto in accuracy performance and alignment rate.

Furthermore, as described in Methods, lr-kallisto is useful for long-read sequencing of single-cell and single-nucleus RNA-seq libraries when coupled with tools designed for barcode discovery [32,38]. Furthermore, lr-kallisto is compatible with translated pseudoalignment, which can be useful for detection of viruses [39].

Finally, in this work we have focused on pseudoalignment and quantification, demonstrating lr-kallisto's significant decrease in walltime and increase in memory efficiency compared with existing long read tools (Table A in S1 Text). However, lr-kallisto can also be used, in principle, for transcript discovery. In particular, reads that do not pseudoalign with lr-kallisto can be assembled to construct contigs from unannotated, or incompletely annotated, transcripts. While we have not completed our investigation and benchmarking of this approach, the pseudoalignment algorithm and distinguishing flanking $k$-mers combine to allow filtering of unmapped reads that do not fit within the annotated model set of transcripts.

## Appendix

## Methods

### Ethics statement

The animal experiments were reviewed and approved by UC Irvine's Institutional Animal Care and Use Committee (IACUC), protocol AUP-21-106, "Mouse genomic variation at single cell resolution".

**lr-kallisto.** Many approaches have been applied to RNA-seq quantification from classical alignment approaches to pseudoalignment paired with likelihoods and expectation-maximization (EM). Due to its speed, efficiency, and accuracy, pseudoalignment with likelihoods and EM has been widely adopted for the mapping of short read RNA-seq. However, for long-read RNA-seq, minimap2 has become the standard for aligning long-reads. Minimap2 follows the standard genome alignment methodology of seed-chain-align [20]. It creates a reference index in the form of hashing minimizers into keys for a reference hash table storing the list of genomic/transcriptomic locations of the minimizer. For each read, minimap2 uses read minimizers as seeds matching these to the reference hash table and identifies the optimal collinear chain(s) of matches. While this method is accurate and has been developed to be highly efficient for the alignment strategy used, it is still time and resource expensive with high memory storage demands.

lr-kallisto, building on the existing framework of kallisto and adapting the pseudoalignment and expectation-maximization algorithm for long-reads, gives an accurate, fast, and low resource solution for mapping long-reads. The main technical

PLOS Computational Biology

challenge of long-reads lies in the higher sequencing error rates, though others include the differing rates of substitutions, deletions, and insertions between long-read sequencing technologies, sequencing length, repetitive regions, and concatemers. To address the challenge of higher sequencing error, different methods, including minimap2 [20], uLTRA [40], and STAR [41] have utilized various approaches to long-read alignment. Minimap2 uses a small $k$-mer size of 14 and 15 for long-reads, while uLTRA employs a two-pass chaining algorithm to improve alignment accuracy. Strobemers have been suggested using fuzzy $k$-mers that allow error tolerance [42]. In lr-kallisto, we, instead, propose a long $k$-mer length and "chaining" pseudoalignment for addressing the challenges of long-read alignment (Figs 4 and 5).

We must address two points: first, that sequencing length and long-read sequencing error rates require a different algorithmic approach to pseudoalignment and, second, the handling of length bias which differs from that of short reads. To address the first, we propose the following algorithm for pseudoalignment and the change of index $k$-mer length to 63, which we discuss after describing the algorithm. Both of these changes take into consideration the sequencing error rate and repetitive regions across genes. While this idea is not a direct implementation of the chaining described in [20], it can be understood in a similar way. Within kallisto's pseudoalignment, a read's transcript compatibility class is determined. For short reads, this is accomplished with a strategy that increases efficiency by checking the transcript compatibility class for the first, middle, and end of $k$-mers in the read if the distance to the end of the contig is longer than the read or the first, middle, and end $k$-mers of the read within the region that is consistent with the contig in the transcriptome de Bruijn graph (T-DBG) (to ensure that the read is consistent with the T-DBG junctions) and then proceeds to the next contig in the read. If they are all the same, these are the only $k$-mers checked, while if they differ a more iterative approach is taken. We then

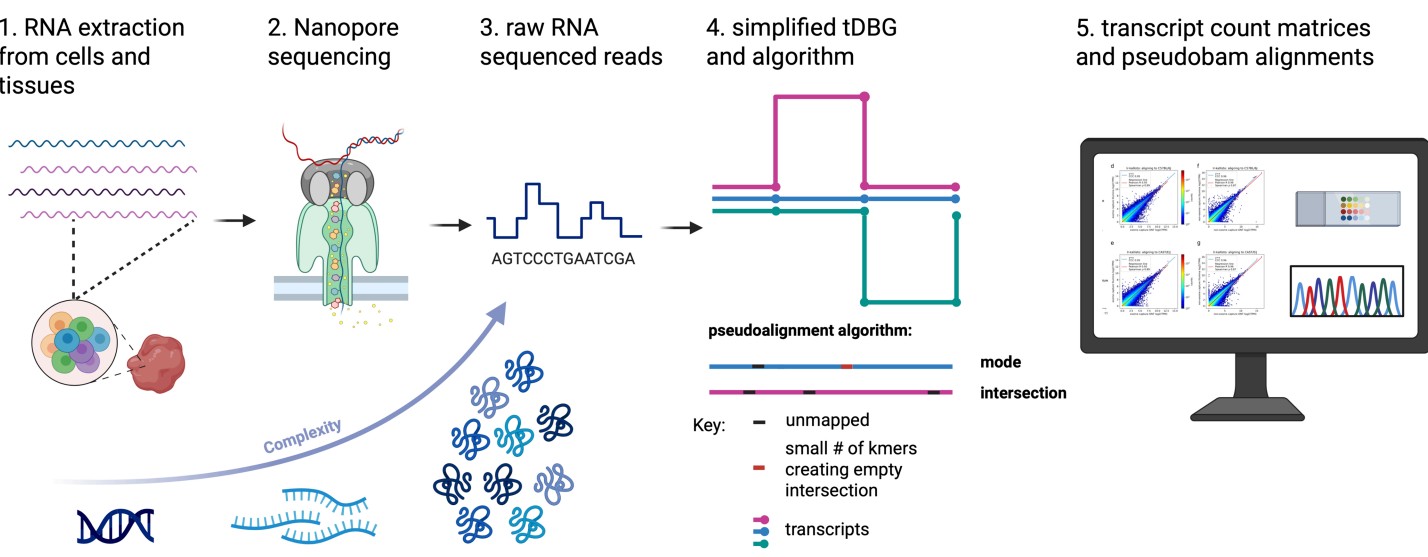

**Fig 4**. **Overview of biosample to lr-kallisto pipeline for long read RNA sequencing.** To study the complexity of life, we can study the genome, transcriptome, and proteome. Through long read sequencing, we can achieve greater insight into both the workings of the genome and the proteome at the individual level and even the functionality of RNA as a molecule. Therefore, improving our ability to analyze long read RNA sequences increases our understanding of biology itself. 1. RNA is extracted from cells and tissues in either single-cell, single-nucleus, or bulk preparation of RNA creating an RNA sequencing library. 2. The RNA sequencing library is then sequenced with either PacBio or Oxford Nanopore Sequencing (Nanopore illustration shown). 3. The raw electrical signal from the nanopore or the raw fluorescent signal from PacBio is then basecalled to create the raw RNA sequenced reads. 4. The raw RNA sequenced reads are input to lr-kallisto outputting both transcriptome quantification of the tissue or single- cells or nuclei as well as the pseudobam alignments for the reads. 5. The analysis and visualization of lr-kallisto's outputs: single-cell or bulk transcript and gene count matrices and pseudobam (pseudoalignments are output in bam format). Created with https://BioRender.com

PLOS Computational Biology

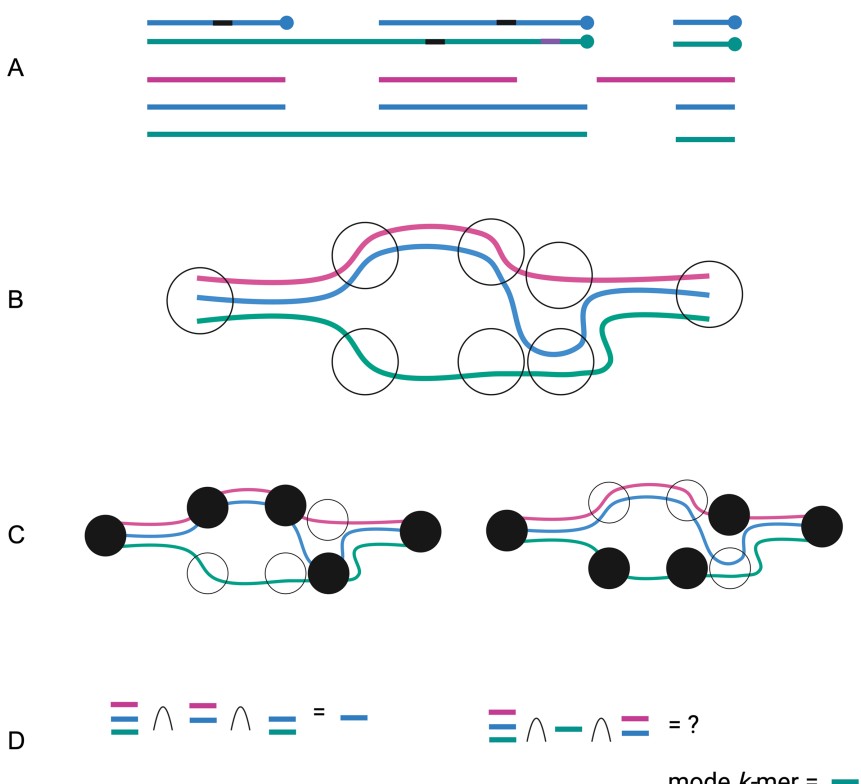

**Fig 5. Overview of lr-kallisto pseudoalignment algorithm.** The input consists of a reference transcriptome and reads from a long read RNA sequencing experiment. (A) An example of two reads (blue and green with unmapping regions (black) and erroneously mapped regions (purple)) and three (pink, blue, and green) overlapping transcripts. (B) An index is constructed by creating the transcriptome de Bruijn Graph (T-DBG) where nodes are $k$-mers, each transcript corresponds to a colored path as shown and the path cover of the transcriptome induces transcript compatibility class (TCC) for each $k$-mer. (C) Conceptually, the k-mers of a read are hashed (black nodes) to find the TCC of a read. (D) The TCC of the read is determined by taking the intersection of the transcript compatibility classes of its constituent $k$-mers, if it exists; otherwise, the mode of the TCCs of the $k$-mers of the read is taken. Created with https://BioRender.com

take the intersection of these transcript compatibility classes. Whereas, in lr-kallisto, if the intersection of transcript compatibility classes (TCCs) a read maps to is empty, we instead take the most often occurring TCC. Moreover, if at least one $k$-mer maps uniquely to a transcript, then we take the most often occurring TCC among mapping $k$-mers that are uniquely mapping to a single transcript. If there are two uniquely mapping regions of the same length within a read to two distinct transcripts, then the read is mapped to the TCC of the first occurring transript in the transcriptome. In the case of the intersection, the intersection can directly be interpreted as the transcript or set of transcripts that the read has the longest combined stretches of compatibility with, since the intersection takes the subset of transcripts that coexist between all $k$-mers with compatible transcripts. However, the intersection may be empty in the case of a variant or error creating an isolated stretch of compatibility with a disjoint transcript compatibility class. Furthermore, in the case that the intersection is empty and the algorithm switches to using the mode of transcript compatibility classes with threshold one, the calculated mode across all transcript compatibility classes that $k$-mers in the read mapped to is the transcript or set of transcripts that again is the "longest chain" of compatibility (Fig 5).

The change of $k$-mer length to 63 was based on empirical evidence showing improved performance over the standard $k$-mer length of 31 for short reads. We found that across long-read technologies and simulations there was an improvement in metrics of Normalized Root Mean Squared Error and Spearman's correlation between lr-kallisto quantifications

and the simulation's ground truth (Figs D–E in S1 Text). In real data (both PacBio and ONT), we observed an increased rate of alignment of reads with a longer *k*-mer length for PacBio sequencing error rate less than 2% and for ONT sequencing error rate less than 10%. Moreover, the longer *k*-mer length improves the quality of mapping *k*-mers making it more probable that the read originates from the transcript compatibility class it maps to. As k increases, the number of distinct *k*-mers also increases, but the number of contigs decreases. This implies that the number of transcripts in a transcript compatibility class decreases on average with increasing length of k. Overall, the complexity of the T-DBG decreases (Fig F in S1 Text), increasing the probability of the read originating from the transcript compatibility class it is mapping to. Furthermore, this also increases the probability of the intersection of equivalence classes being nonempty, which increases the overall mapping rate.

To address the second point, we adapted the effective length, $l_e$, within kallisto to be transcript specific, i.e., defining the effective transcript length, $l_{e_t}$ for a transcript *t* to be:

$$l_{e_t} = \frac{\sum l_{r_t}}{\sum \mathbf{1}_{r_t}} - k$$

where *k* is *k*-mer length, $l_{r_t}$ is the length of a read aligning to transcript *t*, and $\mathbf{1}_{r_t}$ is the boolean function that returns 1 if a read aligns to *t* and 0, otherwise.

We use the first 1 million aligning reads to compute these effective, transcript-specific lengths. The choice of 1 million reads is to be able to compute this expression for every transcript expressed in the data, which is achieved with unordered data and transriptomes of the size of humans and mice. We found that length normalization was effective at low sequencing error rates (< 2%) when sequencing error is uniform, providing a slight improvement in results, and was detrimental to performance at high sequencing error rates and in cases where sequencing error is non-uniform.

We also made lr-kallisto compatible with the d-list option of kallisto [43]. The d-list option produces an index with distinguishing flanking k-mers that serve as certificates to ensure that reads are not mapped erroneously to exons when, in fact, they may have originated from intronic sequence.

Finally, we implemented a change to the expectation-maximization (EM) algorithm for long-reads. In the default option, we initialize transcript abundances to a uniform distribution on the multi-mapping counts with the unique counts for each transcript added to the initialization of a transcript abundance. In the long-read option, we first apportion multi-mapping reads using the EM algorithm starting with a uniform distribution of multi-mapping reads among those mapped to transcripts, and then post EM we add the uniquely mapping counts to each transcript. We found that the latter option works better for the PacBio InDel profile with uniform error in reads in simulations, but that it has reduced performance with real PacBio and ONT reads and simulations based on real data such as with NanoSim's simulations based on profiling of real data. We include the pseudobam option in lr-kallisto, which outputs the pseudoalignments produced by kallisto in bam format, enabling users to analyze where particular reads have aligned, which can also be converted for comparison with genomic coordinates [44].

**Mice and tissue collection.** Mice were housed at the UC Irvine Transgenic Mouse Facility (TMF) in a temperature-controlled pathogen-free room under 12-hour light/dark cycles (lights on at 07:00 hr, off at 19:00 hr). The animal experiments were reviewed and approved by UC Irvine's Institutional Animal Care and Use Committee (IACUC), protocol AUP-21-106, "Mouse genomic variation at single cell resolution". Left cerebral cortex tissues of 10-week-old mice were harvested from 4 C57BL/6J and 4 CAST/EiJ (2 males and 2 females per genotype) between the hours of 09:00 to 13:00. Tissues were stored in 1 mL Bambanker media in cryotubes kept at -80°C until nuclei isolation.

**Purification of nuclei from mouse tissues.** Tissues were thawed in Bambanker media on ice until the tissue could be extracted and lysed using Nuclei Extraction Buffer (Miltenyi Biotec cat. #130-128-024). Using forceps, tissues were transferred to a chilled gentle MACS C Tube (Miltenyi Biotec cat. #130-093-237) with 2 mL Nuclei Extraction Buffer supplemented with 0.2 U/µL RNase Inhibitor (New

England Biolabs cat. M0314L). Nuclei were dissociated from whole tissue using a gentleMACS Octo Dissociator (Miltenyi Biotec cat. #130-095-937). The resulting suspension was filtered through a 70 μm MACS SmartStrainer then a 30 μm strainer (Miltenyi Biotec cat. #130-110-916 and #130-098-458, respectively). Nuclei were resuspended in 3 mL PBS + 7.5% BSA (Life Technologies cat. #15260037) and 0.2 U/μL RNase inhibitor for manual counting using a hemocytometer and DAPI stain (Thermo Fisher cat. #R37606).

**Nuclei fixation.** After counting, 4 million nuclei per sample were fixed using Parse Biosciences' Nuclei Fixation Kit v2 (cat. #ECF2003), following the manufacturer's protocol. Briefly, nuclei were incubated in fixation solution for 10 minutes on ice, followed by permeabilization for 3 minutes on ice. The reaction was quenched, then nuclei were centrifuged and resuspended in 300 μL Nuclei Buffer (Parse Biosciences cat. #ECF2003) for a final count. DMSO (Parse Biosciences cat. #ECF2003) was added before freezing fixed nuclei at -80°C in a Mr. Frosty (Sigma-Aldrich cat. #635639).

**Split-seq experimental protocol.** Nuclei were barcoded using Parse Biosciences' WT Kit v2 (cat. #ECW02030), following the manufacturer's protocol. Fixed, frozen nuclei were thawed in a 37°C water bath and added to the Round 1 reverse transcription barcoding plate at 19,500 nuclei per well, with alternating columns in rows A and C containing C57BL/6J males and females and rows B and D containing CAST/EiJ males and females. In situ reverse transcription (RT) and annealing of barcode 1 + linker was performed using a thermocycler (Bio-Rad T100, cat. #1861096). After RT, nuclei were pooled and distributed in 96 wells of the Round 2 ligation barcoding plate for the in situ barcode 2 + linker ligation. After Round 2 ligation, nuclei were pooled and redistributed into 96 wells of the Round 3 ligation barcoding plate for the in situ barcode 3 + UMI + Illumina adapter ligation. Finally, nuclei were counted using a hemocytometer and distributed into 8 subpools of 13,000 nuclei. The nuclei in each subpool were lysed and cDNA was purified using AMPure XP beads (Beckman Coulter cat. #A63881), then the barcoded cDNA underwent template switching and amplification. Importantly, for two subpools ("13G" and "13H") we increased the number of PCR cycles to 13 cycles from 12, and increased the extension time from 3 minutes to 13 minutes in order to increase the yield of full-length barcoded cDNA. cDNA from one of the subpools ("13G") also received exome capture treatment using Parse Biosciences' Custom Gene Capture Kit (cat. #GCE1001) and a Mouse Exome Panel (Twist Bioscience, cat. #102036). 1 μg of cDNA was hybridized with a blocker solution to block repetitive sequences, then hybridized with the exome panel overnight. Captured molecules were purified using Streptavidin beads, then amplified again using the cDNA amplification reagents from the WT Kit v2 (Parse Biosciences cat. #ECW02030). The cDNA for all 8 subpools were cleaned using AMPure XP beads and quality checked using an Agilent Bioanalyzer before proceeding to Illumina and Nanopore library preparation. All 8 subpools were fragmented, size-selected using AMPure XP beads, and Illumina adapters were ligated. The cDNA fragments were cleaned again using beads and amplified, adding the fourth barcode and P5/P7 adapters, followed by size selection and quality checking with a Bioanalyzer. Libraries were sequenced with two runs of the Illumina NextSeq 2000 sequencer with P3 200 cycles kits (1.1 billion reads) and paired-end run configuration 140/86/6/0. Libraries with 5% PhiX spike-in were loaded at 1000 pM for one run and 1100 pM for the second run and sequenced to an average depth of 301 million reads per library.

**Long-read-split-seq experimental protocol and base calling.** Nuclei were barcoded and cDNA was purified as specified in the previous section. LR-Split-seq libraries were generated using an input of 200 fmol from the amplified, barcoded Split-seq cDNA before fragmentation (section 2 of the Split-seq protocol). Libraries were built using Oxford Nanopore Technologies Ligation Sequencing Kit (SQK-LSK114) and NEBNext Companion Module for Oxford Nanopore Technologies Ligation Sequencing (E7180L). The Short Fragment Buffer (SFB) from the Ligation Sequencing Kit (SQK-LSK114) during the second wash step. Libraries were loaded on R10.4.1 flowcells (FLO-PRO114M, FLO-MIN114) with an input of 20 fmol and 12 fmol, respectively. Sequencing was performed on the GridION and PromethION 2 Solo instruments using the MinKNOW software.

Bases were called from reads with Oxford Nanopore base-calling software Dorado v0.5.0 (https://github.com/nanoporetech/dorado) in super-accurate mode using config file dna_r10.4.1_e8.2_400bps_sup@v4.1.0 for both the exome capture and non-exome capture data, as well as the MinION and PromethION data.

**Long-read-split-seq preprocessing and quantification with splitcode and lr-kallisto.** We first used splitcode to find barcodes and umis using linkers and reverse complements of linkers, allowing a total of 3 errors in linkers. We then used a custom python script to reverse the order of barcodes extracted from reverse strand to be in the same order as forward strand barcodes. Subsequently, we apply splitcode to combine and split randO and polyT barcodes from round 1 of Split-Seq barcoding, allowing 1 substitution or indel per barcode, 39,027,314 out of 105,591,654 raw reads passed this pipeline. We then use lr-kallisto to pseudoalign and quantify the resulting reads; 22,197,716 of the reads pseudoalign. We performed QC with a 500 UMI threshold per nuclei and filtered to genes present in at least 100 cells.

**Error rate estimation.** Error rates for the PacBio dataset [34] were calculated by analyzing a subsample of 1/8th of the reads using the NanoSim read characterization module with the command 'read_analysis.py transcriptome -i *fastq* -rg references/genome.fa -rt references/transcriptome.fa -annot references/annotations.gtf -t 8 -o output_folder'. Error rates for the LRGASP datasets were also calculated this way, without need for subsampling.

**Benchmarking and comparisons.** In benchmarks and comparisons of programs, we used Bambu v3.4.1, IsoQuant v3.3.0, and Oarfish v0.5.1. For the HCT116 data we also ran Oarfish 0.3.1 so as to be able to make a direct comparison with the results of [16]. We ran Oarfish according to the scripts at https://github.com/COMBINE-lab/lr_quant_benchmarks/blob/0b89465420250d3511044fdc3d988a320aba73c6/snakemake_rules/isoquant_sim_data/alignment/alignment_transcriptome/align.snk and https://github.com/COMBINE-lab/lr_quant_benchmarks/blob/0b89465420250d351 1044fdc3d988a320aba73c6/snakemake_rules/isoquant_sim_data/quantification/oarfish_quant/quant.snk. In a previous version of this preprint [45], Oarfish v0.3.1 and v0.4.0 were used and the simulation data was run with SAMtools sort as in [46]. This appears to have resulted in overcounting that degraded Oarfish's performance.

**Data simulation.** The simulation details for SIRV Set 4, where we generate error free reads uniformly expressed across isoforms in SIRV Set 4, are contained in the code for Fig 2 in the GitHub repo https://github.com/pachterlab/LSRRSRLFKOTWMWMP_2024. To see simulation details for Fig 3, see section Data Simulation in [17], which describes in detail the simulation steps used starting with IsoSeqSim and NanoSim as the base simulators and using modifications to NanoSim to better preserve real ONT characteristics. For simulations presented in Fig Ea in S1 Text, we used a custom simulator based solely on error profiles, using ONT error profile of 38.5% of errors are deletions, 38.5% of errors are substitutions, and 23% of errors are insertions and PacBio error profile of 24.5% of errors are deletions, 52.4% of errors are substitutions, and 23.1% of errors are insertions with uniform error distribution within the read, which is full-length, available in the GitHub repo https://github.com/pachterlab/LSRRSRLFKOTWMWMP_2024 with details included with the upload at https://zenodo.org/records/11201284. For simulations presented in Fig Eb-c in S1 Text, we include NanoSim simulation details with the simulated data deposited at https://zenodo.org/records/11201284.

## Supporting information

**S1 Text. Fig A.** Motivation: Comparison of `kallisto` vs `lr-kallisto` on PacBio 1.4% error simulation. **Fig Bi.** Comparison of the percent of reads mapping as spliced vs. unspliced with and without exome capture. **Fig Bii.** Quantifications of C57BL/6J exome capture samples using Bambu, IsoQuant, and Oarfish.
**Fig Biii.** Runtime performance comparisons for `lr-kallisto`, IsoQuant, Bambu, and Oarfish.
**Fig Biv.**

  **I.** (i) Venn diagram of barcodes in ONT and Illumina. (ii) Number of ONT UMI/nucleus vs. Spearman correlation between ONT and Illumina single-nucleus gene-level counts.

  **II.** (i) Venn diagram of barcodes in Illumina random oligo and Illumina poly dT. (ii) Number of random oligo UMI/nucleus vs. Spearman correlation between Illumina priming methods.

  **III.** (i) Venn diagram of barcodes in ONT random oligo and ONT poly dT. (ii) Number of random oligo UMI/nucleus vs. Spearman correlation between ONT priming methods.

**Fig Bv.** Contrast of non-exome vs. exome capture in Illumina and ONT datasets.

**Fig Bvi.** Contrast of priming methods in exome capture comparing Illumina vs. ONT data.

**Fig Ci.** Performance of `lr-kallisto` on ONT-sequenced direct cDNA libraries from HCT116 cell line using Oarfish v0.3.1 (A) and v0.5.1 (B).

**Fig Cii.** Comparison of `lr-kallisto` on ONT-sequenced HCT116 libraries using directRNA and direct cDNA, comparing replicates.

**Fig Ciii.** Evaluation of Bambu, IsoQuant, `lr-kallisto`, and Oarfish on mouse cortex high-depth PacBio data. Created with https://BioRender.com

**Fig D.** Evaluation of `lr-kallisto`, Bambu, IsoQuant, and Oarfish using LRGASP Challenge 2 metrics in Mouse ES cells.

**Fig E.**

  (a) Benchmarks of Bambu, IsoQuant, `lr-kallisto`, and Oarfish across simulation error rates.

  (b) Performance on all annotated transcripts at ONT 11.2% error.

  (c) Performance at ONT 15.2% error.

**Fig F.** Transcript de Bruijn Graph Bandage plots from `lr-kallisto`.

**Table A.** Comparison of tools on memory usage, runtime, percent of aligned and uniquely aligned reads, and total read counts across multiple long-read quantification pipelines.

(PDF)

## Acknowledgments

We thank Zahra Zare Jousheghani, Noor Pratap Singh, and Rob Patro for comments on consistency and version control following the first version of this manuscript on bioRxiv.

## Author contributions

**Conceptualization:** Rebekah K. Loving, Lior Pachter.

**Data curation:** Rebekah K. Loving, Fairlie Reese, Elisabeth Rebboah, Jasmine Sakr, Narges Rezaie, Heidi Y. Liang, Ghassan Filimban, Shimako Kawauchi, Diane Trout, Brian A. Williams, Grant R. MacGregor, Barbara J. Wold, Ali Mortazavi.

**Formal analysis:** Rebekah K. Loving, Lior Pachter.

**Funding acquisition:** Rebekah K. Loving.

**Investigation:** Rebekah K. Loving.

**Methodology:** Rebekah K. Loving.

**Project administration:** Lior Pachter.

**Resources:** Lior Pachter.

**Software:** Rebekah K. Loving, Delaney K. Sullivan, A. Sina Booeshaghi, Páll Melsted.

**Supervision:** Barbara J. Wold, Lior Pachter.

**Validation:** Rebekah K. Loving, Conrad Oakes, Diane Trout.

**Visualization:** Rebekah K. Loving, Fairlie Reese, Conrad Oakes.

**Writing – original draft:** Rebekah K. Loving, Lior Pachter.

**Writing – review & editing:** Rebekah K. Loving, Delaney K. Sullivan, Fairlie Reese, Conrad Oakes, Barbara J. Wold, Ali Mortazavi, Lior Pachter.

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
