## [Decision Letter · Decision Letter 0]

20 Feb 2025

PCOMPBIOL-D-24-02181

Long-read sequencing transcriptome quantification with lr-kallisto

PLOS Computational Biology

Dear Dr. Pachter,

Thank you for submitting your manuscript to PLOS Computational Biology. After careful consideration, we feel that it has merit but does not fully meet PLOS Computational Biology's publication criteria as it currently stands. Therefore, we invite you to submit a revised version of the manuscript that addresses the points raised during the review process.

Please submit your revised manuscript within 60 days Apr 22 2025 11:59PM. If you will need more time than this to complete your revisions, please reply to this message or contact the journal office at ploscompbiol@plos.org. Please include the following items when submitting your revised manuscript:

We look forward to receiving your revised manuscript.

Kind regards,

Tosif Ahamed

Guest Editor

PLOS Computational Biology

Shihua Zhang

Section Editor

PLOS Computational Biology

**Journal Requirements:**

2) Please make sure that the section heading level (Abstract) is clearly indicated in the manuscript text. An outline of the required sections can be consulted in our submission guidelines here:

3) Thank you for including an Ethics Statement for your study. Please include:

i) The full name(s) of the Institutional Review Board(s) or Ethics Committee(s).

5) We have noticed that you have uploaded Supporting Information files, but you have not included a list of legends. Please add a full list of legends for your Supporting Information files after the references list.

Potential Copyright Issues:

i) Figure 1a. Please confirm whether you drew the images / clip-art within the figure panels by hand. If you did not draw the images, please provide (a) a link to the source of the images or icons and their license / terms of use; or (b) written permission from the copyright holder to publish the images or icons under our CC BY 4.0 license. Alternatively, you may replace the images with open source alternatives. See these open source resources you may use to replace images / clip-art:

7) Please amend your detailed Financial Disclosure statement. This is published with the article. It must therefore be completed in full sentences and contain the exact wording you wish to be published.

**Reviewers' comments:**

Reviewer's Responses to Questions

Reviewer #1: The paper by Loving et al proposed lr-kallisto, an updated version of Kallisto specific for long-read transcriptome quantification, which is relevant and useful in the context of fast growing interest in long read RNA sequencing technology. The authors evaluated the performance of lr-kallisto using multiple real datasets, and simulated datasets, and benchmarked lr-kallisto against other methods that are mostly used for long read quantification. In addition, the authors showed an application case of lr-kallisto on single-nuclei samples. In general, the manuscript is relatively clearly written with well presented results, except for some aspects that would need to be addressed or changed.

See below for my comments in details:

Major comments:

1. Application on single-nuclei RNA-Seq samples is under-presented in several ways. Firstly, the conclusion in the first line of the paragraph (page 4 lines 69-70) mentions that efficient performance on single-cell/single-nuclei RNA-seq datasets, i.e., “is not only faster than other tools, but also benefits from the low-memory requirements of kallisto (23, 28).” is not well supported, citations 23, 28 are using kallisto, instead of lr-kallisto. Secondly, results presented in Supplementary Fig 2a, failed to highlight the role of lr-kallisto in this case, where gene level counts can be obtained easily with existing other methods, or even short read single-cell methods like Seurat. The authors should present how transcript level counts like, if possible, with comparison of other methods. Thirdly, the results on barcode seem distracting as it is mainly dependent on another method (splitcode), for better focus, the results on this can be shrinked.

2. The additional value of exome capture is now under-demonstrated and confuses the focus of this paper. The main value of exome capture is probably more relevant to single-nuclei RNA-seq samples, which can be better presented to highlight beneficial impact of exome capture for single-nuclei samples, by showing the results as presented in Fig 1 b-c, Supplementary Fig. 1a for each nuclei, instead of bulk. Another thing is the analysis regarding random oglio vs 3’ primer design needs more investigation. In particular, for page 4 lines 79 to 80 where between exome capture and non-exome capture, CCC is not as high as among exome capture ONT and Illumina, or non-exome capture ONT and Illumina, and then the authors concludes the distorting results suspected from a mix of 3’-end and randomly primed sequencing. This conclusion can be easily proved by repeating the analysis for 3’ end only reads, and randomly primed reads.

3. Page 4 line 65-66 “In addition to being more accurate than other methods, lr-kallisto is also more computationally efficient (Supplementary Fig. 1c)” The comparison is not fair in the sense where alignments in each other method by itself is useful and can be used for downstream analysis. I,.e., for lr-kallisto, minimap2 still needs to be processed separately. A more fair comparison would be excluding minimap2 timing from these methods as they can use alignments immediately. My another concern for this comparison is that mouse data is rather small, a benchmark on human dataset if possible would be very helpful to show how lr-kallisto scales up to big datasets.

4. The manuscript presentation can be improved significantly in terms of figure presentation for supplementary figures, and also method description for lr-kallisto.

Minor comments:

Supplement Fig 1b. The number of points N is missing, and it seems at first glimpse that IsoQuant have much less points. It would help if authors can explain why is that so?

Page 4 lines 71-72 “100% of barcodes from the ONT reads that passed filtering were also found in Illumina sequenced reads” 100% of ONT barcodes found in illumina vs only half of Illumina barcodes found in ONT, why is so? Is it because of sequencing depth, if increasing ONT sequencing depth, would the results remain the same?

Page 4 lines 73-74 “Increased UMI depthper nucleus yields higher Spearman correlations, indicating that with deeper sequencing depth, short and long read correlations will only improve” What is the range of sequencing depth corresponding to the increasing UMI depth? What each point represent here?

Supplementary figure 1e: for illumina reads, is it kallisto instead here? The current representation is a bit confusing.

Page 4 lines 83-85, what is the estimation method for Illumina dataset? Based on the y-axis label, it seems kallisto Illumina results is used for comparison against lr-kallisto results. But for comparison against oarfish results, the method for Illumina is unclear. Please specify the method used here and explain further on the potential impact of method for Illumina data.

Page 4 lines 86 to 87, the comparison between direct cDNA and direct RNA against Illumina results differ between replicate 3 and replicate 4, where for replicate 4, direct cDNA concordants are better than direct RNA, while for replicate 3, direct RNA concordants are better. Hence, the conclusion is not accurately described. Would recommend to rephrase.

Suppl Fig 2c. The method for Illumina used here is unclear and might impact how the results can be interpreted.

Page 11 line 281: “the length bias in sampling longer transcripts less times”is it the opposite instead?

Page 12 line 313: “We use the first 1 million aligning reads to compute these effective, transcript-specific lengths”Why use 1st million aligning reads? Would it be biased? How representative is it? This needs to be evaluated.

Reviewer #2: Loving et al. present lr-kallisto, a modification of the kallisto pseudomapper tailored for long-read PacBio and Nanopore data. They compare lr-kallisto with other state-of-the-art quantification tools using multiple datasets, including simulated data. They demonstrate the tool's performance on single-nucleus LRS RNA-seq, both with and without exome capture. They claim that lr-kallisto outperforms other tools in quantification accuracy when compared with Illumina data. Additionally, lr-kallisto is faster.

This study makes a significant contribution to the field of long-read RNA-seq quantification, where faster tools are necessary due to the increasing throughput of sequencing technologies. The manuscript shows that lr-kallisto performs comparably or better than other quantification tools. However, I would like to see further details on how well lr-kallisto quantifies "difficult cases"—specifically, when alternative isoforms are expressed simultaneously and when reads are incomplete. This is crucial because the primary advantage of long-read RNA-seq is its ability to detect alternative isoforms, and tools utilizing these data should address this capability explicitly.

Specific Comments:

1. The authors show that lr-kallisto exhibits a stronger correlation with Illumina data than other tools. However, I wonder if short-read quantification was performed using kallisto. If so, this might introduce bias due to the expected high concordance between kallisto-based methods. This potential bias should be acknowledged in the discussion.

2. As a pseudomapper, lr-kallisto should not only be compared with quantification tools but also with mappers such as minimap2 and uLTRA. While I understand that direct comparisons might be challenging because kallisto does not perform conventional mapping, certain analyses could still be informative. For example, a key distinction between minimap2 and uLTRA is that uLTRA improves the identification of small exons. How well does lr-kallisto handle this scenario, particularly when alternative isoforms are expressed? This could be evaluated using simulated data.

3. A common issue in long-read RNA-seq is the presence of truncated reads, particularly at the 5' end, which complicates the quantification of transcripts with alternative transcription start sites (TSS). How does lr-kallisto perform in such cases? Can the algorithm accurately quantify isoform expression when alternative TSSs are present? Again, this could be tested using simulated data.

4. The parameters used for generating simulated data should be described in greater detail. Specifically, the authors should clarify how multiple isoforms of the same gene were simulated, whether expression values were kept constant or varied and whether the simulated data allowed for the assessment of minor alternative isoforms.

5. The authors suggest that lr-kallisto, despite being reference-based, could be used to identify novel transcripts by assembling reads that are not pseudomapped. This claim seems overstated and is not demonstrated in the manuscript. However, it would be interesting to assess how many reads are not used for quantification and how this compares to other tools. While some relevant data is present, a clear table summarizing these numbers would be beneficial. This table should ideally include the percentage of unmapped reads for each tool including nresults from Illumina-based kallisto for reference.

6. The authors use data that includes spike-ins, which provide a ground-truth dataset, yet the manuscript does not present quantification results for these spike-ins. Including this analysis would strengthen the study.

7. The performance of pseudomappers relies heavily on the availability of a well-annotated transcriptome, which is not available for many species. Moreover, long-read RNA-seq has consistently revealed thousands of novel transcripts in nearly every experiment. Excluding unannotated transcripts from quantification inherently affects the performance of these tools. It would be valuable to see an analysis using partially annotated transcriptomes, possibly through simulation. While this might be beyond the current scope of the manuscript, the authors should discuss these considerations more thoroughly.

Additional Remarks:

• The description of the lr-kallisto algorithm is difficult to follow, especially for readers unfamiliar with kallisto. I strongly recommend adding a graphical figure to illustrate the algorithm.

• The manuscript states that when the TCC intersection of a read is empty, the most frequently occurring TCC is selected. However, what happens in cases of ties? This should be clarified.

• Line 302: Please clarify which ground truth is being referenced.

• Line 363: Indicate the exact number of spike-in reads used in the analysis.

Reviewer #3: This manuscript describes an update in the RNA-seq quantification software kallisto to enable quantifying transcriptomes from long-read sequencing data. The described algorithm seems to properly address the problem of performing fast and accurate pseudoalignments with long-read higher error rates. The benchmarking results show a very good agreement with Illumina data and a competitive performance with other existing tools, while providing the very fast execution times of a pseudoalignment-based approach. In summary, this is a useful addition to the kallisto pipeline and to the quantification toolkit for long-read sequencing data. I only have a few minor comments that I'm listing below.

Minor comments:

* There are probably too many panels in figures 1 and 3, which makes axes labels in scatter plots too small to be read comfortably. In fact, the label "counts" on the right-axis of panels d and e of figure 1 is cropped. a solution would be to produce more figures with fewer panels.

* Something I do not understand from the color map representing the counts in the scatter plots of figures 1 and 3 is that, according to that color map, blue corresponds to lower counts and red to higher counts. however, blue appears associated with higher log2 TPM values and red to lower ones. Should not this be the other way around?

* In page 7, line 152, "comptible" -> "compatible"

* In page 12, after line 312, the displayed equation has no number and its notation is somewhat unclear, I guess 'l_{reads aligning to t}' refers to the length of the reads, but this is not explicitly mentioned and the summation has no index. The term in the denominator uses also quite a non-standard mathematical notation. I'd recommend the authors to use a more standard notation with single letters, describing them in the main text.

* In page 12, it is unclear to me what it means that an option has "reduced performance in the wild"?

**Have the authors made all data and (if applicable) computational code underlying the findings in their manuscript fully available?**

Reviewer #1: Yes

Reviewer #2: Yes

Reviewer #3: Yes

PLOS authors have the option to publish the peer review history of their article (what does this mean?). If published, this will include your full peer review and any attached files.

Reviewer #1: No

Reviewer #2: No

Reviewer #3: No

**Figure resubmission:**
---

## [Decision Letter · Decision Letter 1]

21 Jul 2025

PCOMPBIOL-D-24-02181R1

Long-read sequencing transcriptome quantification with lr-kallisto

PLOS Computational Biology

Dear Dr. Pachter,

Thank you for submitting your revised manuscript to PLOS Computational Biology. We have received feedback from our reviewers. While Reviewer #3 has confirmed that all their previous concerns have been adequately addressed, Reviewer #1 has raised some remaining points that require attention, particularly point 2, where Reviewer #1 has expressed confusion regarding the barcode overlap between ONT and Illumina platforms.

Therefore, we invite you to submit a revised version of the manuscript that addresses the points raised during the review process.

Please submit your revised manuscript within 30 days Sep 20 2025 11:59PM. If you will need more time than this to complete your revisions, please reply to this message or contact the journal office at ploscompbiol@plos.org. Please include the following items when submitting your revised manuscript:

We look forward to receiving your revised manuscript.

Kind regards,

Tosif Ahamed

Guest Editor

PLOS Computational Biology

Shihua Zhang

Section Editor

PLOS Computational Biology

**Journal Requirements:**

1) Thank you for providing your Data Availability Statement. We noted that these links  "https://github.com/ pachterlab/kallisto" and "https://github.com/pachterlab/ LSRRSRLFKOTWMWMP_2024" reach 404 error pages. Please amend them to working links.

2) Please amend your detailed Financial Disclosure statement. This is published with the article. It must therefore be completed in full sentences and contain the exact wording you wish to be published.

1) State what role the funders took in the study. If the funders had no role in your study, please state: "The funders had no role in study design, data collection and analysis, decision to publish, or preparation of the manuscript.".

**Reviewers' comments:**

Reviewer's Responses to Questions

Reviewer #1: I appreciate authors’ efforts in addressing my questions, most of them are addressed now with some minor points remained.

1. Supplementary Table 1: please explain what is d-list. Also, for pseudobam, more details need to be mentioned in main text. For now, this is only mentioned in Figure legend. How comparable it is with minimap2 bam files, also uLTRA as mentioned by another reviewer.

2. The barcode overlap between ONT and Illumina is still a bit confusing to me as I am unclear why would near 100% overlap is expected, given that less than half of the barcodes in Illumina can be found in ONT. Why would ONT barcodes all found in Illumina? I would suggest authors to either soften their speculation on this, or provide additional support for the current speculation.

Reviewer #3: The authors have correctly addressed all the comments in my review.

**Have the authors made all data and (if applicable) computational code underlying the findings in their manuscript fully available?**

Reviewer #1: Yes

Reviewer #3: Yes

PLOS authors have the option to publish the peer review history of their article (what does this mean?). If published, this will include your full peer review and any attached files.

Reviewer #1: No

Reviewer #3: No

**Figure resubmission:**
---

## [Editor Report · Decision Letter 2]

31 Oct 2025

Dear Prof. Pachter,

We are pleased to inform you that your manuscript 'Long-read sequencing transcriptome quantification with lr-kallisto' has been provisionally accepted for publication in PLOS Computational Biology.

Best regards,

Shihua Zhang

Section Editor

PLOS Computational Biology

Shihua Zhang

Section Editor

PLOS Computational Biology

---

## [Editor Report · Acceptance letter]

PCOMPBIOL-D-24-02181R2

Long-read sequencing transcriptome quantification with lr-kallisto

Dear Dr Pachter,

I am pleased to inform you that your manuscript has been formally accepted for publication in PLOS Computational Biology. Your manuscript is now with our production department and you will be notified of the publication date in due course.

With kind regards,

Lilla Horvath
